# China’s Changing Alcohol Market and Need for an Enhanced Policy Response: A Narrative Review

**DOI:** 10.3390/ijerph19105866

**Published:** 2022-05-11

**Authors:** Shiwei Liu, Fulin Huang, Xiaolei Zhu, Suhua Zhou, Xiang Si, Yan Zhao, Yang Liu, Xiaochang Zhang, Sally Casswell

**Affiliations:** 1Tobacco Control Office, Chinese Center for Disease Control and Prevention, Beijing 100050, China; zhijb_huangfl0602@163.com (F.H.); liuyang19948196@163.com (Y.L.); 2Noncommunicable Disease and Aging Health Management Division, Chinese Center for Disease Control and Prevention, Beijing 102206, China; zhuxl@chinacdc.cn (X.Z.); sixiang@chinacdc.cn (X.S.); zhangxc@chinacdc.cn (X.Z.); 3Hubei Provincial Center for Disease Control and Prevention, Wuhan 430079, China; zhousuhua@sina.com; 4Department of Public Health, School of Public Health, Inner Mongolia Medical University, Huhehot 010110, China; zhaoyan8806@foxmail.com; 5Social and Health Outcomes Research and Evaluation (SHORE), SHORE & Whariki Research Centre, College of Health, Massey University, P.O. Box 6137, Victoria Street West, Auckland 1142, New Zealand; s.casswell@massey.ac.nz

**Keywords:** China, alcohol, industry, marketing, consumption, affordability, policy response

## Abstract

This study describes trends in alcohol consumption in the context of an expanding commercial context, current policy responses, and flaws in relation to international best practice for alcohol control in China. We surveyed the literature and other documents in Chinese or English up to December 2020 on policy responses to alcohol consumption and harm, industry structure, and marketing practices in China. Databases searched included PubMed, China National Knowledge Internet, Wanfang Data, Web of Science, and Baidu Scholar. We also scanned the official websites of government organizations and gathered information using snowballing. We analyzed existing alcohol policy against evidence-based, cost-effective policies for reducing alcohol harm. Our findings show that although some restrictive policies have been enacted with potential impacts on alcohol harm, they are not comprehensive, and some are poorly executed. The long history of alcohol use remains an important element in alcohol consumption by the Chinese population. However, alcohol marketing and promotion, ease of access, and affordability have become increasingly prominent. The gaps identified in alcohol policy suggest improved strategies and measures to reduce the harmful use of alcohol are urgently needed in China.

## 1. Introduction

Alcohol use is a major risk factor for the global burden of disease [1]. The Global Burden of Disease Study (GBD) 2019 showed that the disease burden attributable to alcohol use relative to 87 risk factors ranks 9th in population and 6th among men globally. In China, it is 8th in the population and 5th among men. Alcohol use is also an important factor leading to the disability and death of adolescents in China and accounted for 39.1% of adolescent deaths, ranking as the second leading cause of death. Among those aged 10–24, the disability-adjusted life years (DALY) caused by alcohol use is 620,751, ranking as the first cause of DALYs (18.5%) [2].

A global goal has been established to reduce alcohol consumption by 10% by the year 2025 [3]. While alcohol consumption has decreased in Europe over the past three decades, there have been increases in many middle-income countries driven by economic growth and the commercialization of alcohol. China has experienced overall growth over these decades as it moved from a low-income to upper middle-income country. The trend of increased alcohol consumption in China will influence not only its own experience of alcohol harm but will also be a major contributor to the outcome of the global goal.

China has a long history of alcohol use, having evidence of brewing going back to about 9000 years ago [4,5,6], and alcohol use and drinking etiquette are important parts of festivals and daily life, including in traditional medicine [7,8,9]. However, little is known about the important influences of the alcohol industry on alcohol use in China, and about the nature and extent of a public health policy response to prevent alcohol harm.

This narrative review aims to identify the changing nature of alcohol production and trends in alcohol consumption. It also examines China’s policy responses in the context of what are known to be the most cost-effective alcohol policies [10,11] and identifies gaps in policy development and implementation.

## 2. Methods

We surveyed the published literature and publicly available grey literature and documents, in Chinese or English and up to December 2020, on alcohol origins and drinking culture in China; the alcohol industry’s activity in China, including online marketing and promotion; alcohol consumption in China; and the policy response to alcohol use in China. The databases searched included PubMed, China National Knowledge Internet, Wanfang Data, Web of Science, and Baidu Scholar. The official websites of government agencies and other organizations, mainly alcohol-related industry associations, were scanned, and information was also gathered using forward/backward citation searches. Official documents mainly referred to alcohol-related laws, regulations and directives, and guidance from involved agencies.

For our searches, we used variations and combinations of the keywords—alcohol, alcoholic beverage or drink, baijiu, spirits, wine, rice wine, rice beer, yellow wine, beer, origin, culture, industry, production, brewing, advertising, marketing, promotion, prevalence, consumption, tax, control, policy, and control policy—and their Chinese translations. Six of the authors separately retrieved, screened, and reviewed one or two of the abovementioned topics each—alcohol origin, culture, industry, production, marketing, promotion, prevalence, consumption, tax, and policy responses to alcohol use. Any duplicates retrieved from different sources were excluded, as well as unverifiable information from non-official websites or informal publications, or information deemed irrelevant or insignificantly relevant to the proposed theme. One author was responsible for collating all the information from the six authors and carrying out quality checks against the exclusion criteria.

Based on the information on alcohol origins, culture, industry, marketing, promotion, consumption, and policy responses to alcohol use gathered and summarized in the review process, we identified the main challenges and opportunities for alcohol control in China, especially policymaking and implementation.

## 3. Results

### 3.1. Production and Consumption Trends

From the founding of the people’s Republic of China to the early stage of reform and opening up (1949–1985), China’s liquor industry was in an early stage of industrial development. During this period, small workshops began to merge into larger-scale producers [12]. The earliest data available from the WHO [13] show that between 1960 and 1979, per capita adult consumption of alcohol was between 0.5 and 1.5 L of pure alcohol.

Between the “Seventh Five-Year Period” and the “Eighth Five-Year Period” (1986–1995), the system of ownership and control of alcohol production and retail changed. In 1991, the legislative change allowed a move from the planned supply and state-owned monopoly system in order to adapt to the development of a market-oriented economy [14]. The Regulations of the People’s Republic of China on Alcohol Administration changed the alcohol monopoly system to a licensing system, and alcohol sales were allowed to be privatized [14].

The alcohol industry entered a stage of rapid development. The beer industry began to collectivize and scale up. Enterprises consolidated through acquisitions, mergers, and asset restructuring [15,16], and trade associations were formed. During this decade, per capita consumption of recorded alcohol increased from under 3 L per capita to approximately 4 L [13].

During the “Ninth Five-Year Period” (1996–2000), China issued a series of policies regulating the development of the alcohol industry, and most of the producers of spirits and yellow wine, the two predominant beverages, were still small scale brewers with regional sales [17,18]. Consumption per capita of recorded remained between 3 and 4 L [13].

As of 2010, the alcohol industry was effectively consolidated, and the competitive nature of the industry increased. The number of large enterprises with assets above 400 million yuan, annual income above 300 million yuan, and more than 2000 employees increased from 43 in 2010 to 108 in 2014. Competition in the alcohol industry was intense, and to increase their market power, companies engaged in mergers and acquisitions; consolidation increases access to information, technology, and capital [19]. Per capita consumption of recorded alcohol in 2010–2019 was approximately 5 L [13].

#### 3.1.1. Current Alcohol Production and Marketing

Most of the alcohol production companies in China are not under state ownership. In 2019, of the top 174 liquor production enterprises of 200 Chinese liquor brands, 45 (26%) were state-owned enterprises, 114 (66%) private enterprises, 10 (6%) domestic and foreign joint ventures, and 5 (3%) wholly foreign-owned enterprises [20].

Chinese spirits producers are among the largest alcohol producers globally, and the Moutai brand was ranked number one in terms of brand value in 2020 [21]. China’s top four spirits producers dwarf well-known global spirits brands, such as Johnnie Walker [22]. While foreign ownership is still a minority of the industry, some of the largest transnational alcohol corporations producing beer and western spirits are now active in China. ABInBev, the world’s largest alcohol corporation, has operated in China, starting with technological exchange, for more than 30 years [23]. It now markets global brands such as Budweiser and national brands such as Harbin [24].

Many methods of promoting alcohol consumption are employed in China. Alcohol brands are promoted in China through television, magazines, newspapers, radio, and digital media [25,26,27,28,29]; and promotions include free samples, buy one get one free, price reductions, and shopping coupons [30]. Alcohol brands are also promoted through celebrity endorsements, sponsorships, tastings, charity activities, beer festivals [31], and cultural festivals [32,33]. Alcohol brands are marketed in association with tourism, unrelated products, and consumers’ interests [34]; and there is direct sales promotion, promotion through middlemen or agents, and retail promotion in shopping malls, supermarkets, restaurants, and hotels [35,36,37]. Online shopping is a fast-expanding market trend, favored by young people, and the alcohol industry began using e-commerce in 2009 [38].

#### 3.1.2. Trends in Consumption

The World Health Organization’s (WHO) estimate of China’s per capita alcohol consumption in 2016, including unrecorded alcohol, was 7.2 L of pure alcohol, 12.5% higher than the global average (6.4 L). Recorded per capita alcohol consumption showed an overall upward trend, from 1.3 L in 1979 to 5.7 L in 2016, whereas unrecorded alcohol has shown a decrease. It was predicted that by 2020 and 2025, the total per capita alcohol consumption of Chinese people aged 15 and above would be 7.5 and 8.1 L of pure alcohol, respectively [13].

The prevalence of drinking has increased in China. In 1991, 2002, and 2010–2012, the prevalences of alcohol consumption in the past 12 months reported in surveys among adults aged 15+ were 17.94%, 21.0%, and 34.3% respectively [39,40,41]. A national epidemiological survey [42] showed that the prevalence of alcohol consumption in 2018 among Chinese adults aged 18 and over in the past 12 months was 39.8%, much higher in men (60.3%) than women (19.1%), highest among those aged 18–59, and not significantly different in urban areas compared to rural areas. Meanwhile, the rate of harmful use of alcohol was 8.6%, much higher in men (10.7%) than women (1.6%), and higher in rural (10.5%) than urban areas (6.9%). The proportion of occasional heavy drinking was 39.8% (46.8% for men and 17.5% for women).

### 3.2. Policy Responses to Alcohol Use

As of 2020, China had a number of policies in place relevant to the management of alcohol consumption and harm. Some align with the UN best buys, the most cost-effective alcohol policies, which are control of availability, taxation to ensure affordability is controlled, and restriction of marketing [11]. However, in China these are not comprehensive or well enforced.

The availability of alcohol is subject to licensing covering production [43,44]; controlling the quality of the products [45,46]; regulation of distribution, including record registration, operation rules, supervision and management, and legal liability of alcohol wholesale, retail, storage, and transportation companies; and other business activities [47]. However, online delivery has become an important tool of supply of alcohol in China and is not subject to regulation. The policies on alcohol distribution which are in place are poorly implemented, and policies regarding management of the supply of alcohol and of the drinking environment are not implemented [48].

Another effective policy is preventing access to alcohol among younger people. Several laws with a focus on the protection of minors [49], prevention of juvenile delinquency [50], and health promotion [51] stipulate that the minimum age for alcohol purchase and consumption in China is eighteen years old, and set provisions to prevent minors getting drunk, selling alcohol, and drinking alcohol. However, regulation of minors’ access is not well enforced. A survey in 2021 across seven provinces in China indicates that among middle and high school students who had attempted to purchase alcohol in the past 30 days, 82.5% reported they were not refused because they were underage [52].

Taxation is one of the most-cost effective policies for reducing alcohol harm, providing it affects the affordability of the products [53]. In China, both value-added tax (17%) and a specific excise tax are in place. The tax is mainly collected in the production chain, using a compound calculation method of ad valorem and a quantitative levy. For example, an ad valorem tax for baijiu is levied on 20% of sales volume, and the quantitative levy is 0.5 yuan on 500 mL [54]. Rates vary based on the type, production, cost, and use of alcohol. They currently are: 240 yuan per ton for yellow wine, 250 yuan per ton for higher priced beer (ex-factory 3000 yuan and above), 220 yuan per ton lower priced beer (below 3000 yuan), 10% for other alcohol, and 5% for ethyl alcohol. The rate for the specific tax on baijiu is about 12%, accounting for about 80% of all the taxes of alcohol enterprises. In 2015, the alcohol industry paid a total of 102.753 billion yuan in taxes [55]. However, the affordability of alcohol has increased, and a draft law [56] currently under consultation would keep the current tax system framework and tax burden level generally unchanged [56,57,58].

Regulation of the marketing of alcohol is another cost-effective alcohol policy. Laws exist in China which partially regulate marketing via the advertising content, time and frequency, and protection of minors [59,60,61]. However, there is not a comprehensive marketing ban in place, and there is no regulation on new forms of alcohol marketing, such as digital marketing, including promotion of online delivery of alcohol.

Regulations to prevent drink driving have also been shown to be effective at reducing alcohol harm. China sets two BAC levels: ≥20 mg/100 mL is defined as drink driving and ≥80 mg/100 mL as drunk driving. Both drink and drunk driving are penalized [62,63]. Recidivism is subject to higher penalties, and in 2011, drunk driving became a criminal offence leading to imprisonment and a fine [64,65].

## 4. Discussion

This descriptive analysis provides a detailed overview of the alcohol market and alcohol policy in 2021. Previous commentators have identified a relative lack of effective alcohol policy in China as an issue of concern [66,67], and this analysis shows an ongoing need for greater focus on alcohol policy to protect public health, not just to respond to the previous policy gaps but also to meet new challenges, such as digital marketing [68].

While a planned supply and state-owned monopoly system of distribution of alcohol was in place, it was possible for the government to determine the rate at which the market for alcohol expanded. The increased consolidation and competition between private alcohol producers encourages greater expansion of the market. In these circumstances, policies in the form of laws and regulations affecting availability, affordability, and marketing are necessary to minimize harms. Such policies are unpopular with private interests given the impacts on profits, and internationally the engagement of the transnational alcohol producers in the policy process has been shown to have a negative impact on the development of effective policy [69,70,71].

The alcohol industry’s influence is often via their public relations activities, and in this regard it is interesting that ABInBev has chosen China for one of the sites of its global Smart Drinking Campaign which employs celebrities and partnerships with national agencies, often focused on drink driving concerns [23]. Another example, Harbin beer, promotes the globally widely promoted “drink responsibly” message to young drinkers who are identified as a target group for the brand [24]. ABInBev has also contributed to Project Hope in China, a huge government-endorsed education project seeking non-governmental contributions to overcome educational inadequacy in poverty-stricken rural communities, and has committed to building a Hope School when a new brewery is developed [72].

The intensity of current alcohol marketing in China illustrates the drive to expand profits, and the partial restrictions on marketing which are in place are unlikely to have any impact on reducing its power [10]. In a context of the widespread use of digital media in China, the failure to ban the promotion of alcohol on social media and other digital platforms is a major policy gap.

Easy access to alcohol reflects a lack of adequate regulation on the places selling alcohol and hours of sale. As in many countries, the legal minimum age for purchase and consumption of alcohol is not enforced in China, so alcohol ranks highly as a contributor to mortality among adolescents. A major challenge in control of the supply of alcohol facing China is online sales, the value of which rose in China from nearly 4 billion yuan in 2012 to more than 100 billion yuan in 2020 [73].

One of the most cost-effective alcohol policies is a taxation regime which actually affects the affordability of alcohol products. From a public health perspective, it is recommended that tax reflects the potency of the beverage, as this is responsible for the harm but is also adjusted for economic changes so as to maintain the cost to the consumer over time. China’s taxation system reflects the higher potency of spirits to some extent. However, the retail price index of alcohol products has shown a downward trend, indicating that the relevant tax policies have not played an effective role in controlling affordability, nor, therefore, consumption and harm [53]. A draft law under consideration would prevent the alcohol tax being increased in order to reduce harm, despite the recommendation in China’s plan to reduce NCDs to improve tax policy. As the population increases in wealth, if effective policies are not put in place, increases in consumption and harm are anticipated [74].

Laws, regulations, and policies focused on alcohol control do exist in China. However, these could be more closely aligned with the most cost effective alcohol control policies, as outlined as part of the NCD action plan [75,76,77], and be more adequately implemented. The involvement of the private interests in the development of alcohol policy, either international or national [78], will militate against effective policy, the development of which is a government responsibility.

Two further regulations, introduced in 2012 [79] and 2013 [80], had potential impacts on the alcohol market while not being public control measures. These policies set strict regulations on alcohol use by official personnel in public affairs for the purpose of “incorrupt government,” and supervision requirements. These “uncorrupt government” regulations introduced in 2012 and 2013 impacted the alcohol industry in China [80,81], but there is no evidence of an impact on alcohol consumption population wide. There was a substantial effect on the state-owned producer of expensive liquor but relatively little on producers of cheaper distilled liquor. The net profit growth of Kweichow Maotai was 3.6% in the first half of 2013, the slowest growth since 2001 [79]. However, the industry responded by reducing prices and expanding e-commerce and online delivery [82], and the longer term impact of these regulations is not known.

Drink driving legislation is an effective policy to reduce harm from alcohol, and the change in legislation in 2011 to make driving under the influence of alcohol a criminal offense [64,65], which would also result in public officials losing their employment, may have had benefits in decreasing crashes, but this has not been subjected to a well-designed evaluation based on reliable data.

The main efforts to reduce the negative consequences of alcohol use and drunk drinking focus on the dissemination of knowledge of alcohol’s hazards [83], which is not a cost-effective approach. Additionally, there is no specialized office or agency or organization responsible for alcohol control, no corresponding multi-sectoral coordination mechanism, and no specific agency to conduct regular monitoring of alcohol use [3]. China does not yet have a comprehensive monitoring system for alcohol consumption and its related diseases. The existing policies on alcohol control have not set specific targets for reducing the drinking rate and amount per capita.

Consumption of alcohol has increased in China in the context of privatization of alcohol production and increased disposable income among the population. However, the prevalence of drinking in China, particularly among women, remains lower than in many high-income countries. A future increase in prevalence is likely to reflect an increase in drinking among women and younger people, as has been seen elsewhere [84]. How fast the prevalence increases will largely reflect economic growth unless more effective policies are put in place.

## 5. Limitations

Narrative reviews are not systematic, so some relevant materials could have been missed, but the authors’ goal was to provide a narrative thread on the topic rather than a comprehensive, catch-all review. A second limitation is that only official information that is publicly available could be accessed.

## 6. Conclusions

The changing nature of alcohol production in China has resulted in consolidated and highly competitive companies. While most production is by companies with domestic ownership, transnational corporations are expanding their role. In the context of consolidation and competition, widespread supply, ensuring easy availability, and marketing—encouraging increased use, new products, and drinking contexts—characterize the alcohol industry. China’s policy responses in terms of the most cost effective are in urgent need of reform. Marketing legislation does not adequately protect the population, and restrictions on availability are not present or not enforced. Alcohol has become more affordable. To restrain consumption increases and minimize alcohol-induced harms in the future, these are areas which need urgent attention.

## Data Availability

Not applicable.

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
