# Peer review of "China’s Changing Alcohol Market and Need for an Enhanced Policy Response: A Narrative Review"

_ijerph, 2022, doi:10.3390/ijerph19105866_

Round 1

Reviewer 1 Report

Very good literature review and synthesis of the results. I have just a few comments and suggestions:

(1) METHODS, page 2, lines 64-65: The databases searched included PubMed and Web of Science. There are other databases that may have produced additional literature and policies. These include: Article Plus; Cochrane Library; Pro Quest; WorldCat; PsychINFO; Social Services Abstracts; and various conference proceedings (e.g. International Council on Alcohol, Drugs and Traffic Safety (ICADTS); Association for the Advancement of Automotive Medicine; Research Society on Alcoholism). I would suggest you search these sources. 

(2) There is no mention of Responsible Beverage Service (RBS) training and enforcement. Is there no training of alcohol servers in China?

(3) There are several laws that help reduce underage drinking such as Dram Shop and Social Host Liability. Not in China?

(4) Screening, Brief Intervention and Referral (SBIR) is also effective in reducing alcohol harm. Not in China?  

Author Response

Very good literature review and synthesis of the results. I have just a few comments and suggestions:

  • METHODS, page 2, lines 64-65: The databases searched included PubMed and Web of Science. There are other databases that may have produced additional literature and policies. These include: Article Plus; Cochrane Library; Pro Quest; WorldCat; PsychINFO; Social Services Abstracts; and various conference proceedings (e.g. International Council on Alcohol, Drugs and Traffic Safety (ICADTS); Association for the Advancement of Automotive Medicine; Research Society on Alcoholism). I would suggest you search these sources. 

We believe the search databases we used meet the need for a narrative review (see below for more information regarding narrative reviews)  and given the lack of research on China alcohol-related policy we have reached saturation with the current searches.

  • There is no mention of Responsible Beverage Service (RBS) training and enforcement. Is there no training of alcohol servers in China?

This is not found to be an effective intervention (eg Babor et al, 2010) and therefore was not included in the paper.

  • There are several laws that help reduce underage drinking such as Dram Shop and Social Host Liability. Not in China?

These do not exist in China.

  • Screening, Brief Intervention and Referral (SBIR) is also effective in reducing alcohol harm. Not in China?  

This approach has some effectiveness at the individual level but less at the population level.  Appropriate technologies to provide rapid alcohol dependence screening and brief abstinence interventions are not available in China.

Reviewer 2 Report

The authors analyze of alcohol consumption in the People's Republic of China is based on literature on the subject available in English and Chinese prior to December 2020. The article is interesting, drawn up in an orderly fashion. However, it focuses only on data on alcohol production and consumption in China by people of different age groups, and on government policies aimed at reducing this consumption. The authors are convinced that the reason for the increase in alcohol consumption is the abandonment of state monopoly, characteristic for communist countries. However, they do not address the reasons for the increase in alcohol consumption other than its free market availability and promotion. One could, however, ask about non-market determinants of the growing alcohol consumption, e.g. cultural, political, psychological, etc. This means that they associate the possible decrease in alcohol consumption only with an increase in taxes, an expansion of legal restrictions, and increased state control and monitoring. This does not seem to be a legitimate conclusion.
The authors report that according to the World Health Organization, per capita alcohol consumption in China in 2016 was 12.5% higher than the global average (142-143), and then go on to state that it is lower than in many highly developed countries (283-285). This would seem to require additional commentary.

Author Response

Comments and Suggestions for Authors

The authors analyze of alcohol consumption in the People's Republic of China is based on literature on the subject available in English and Chinese prior to December 2020. The article is interesting, drawn up in an orderly fashion. However, it focuses only on data on alcohol production and consumption in China by people of different age groups, and on government policies aimed at reducing this consumption. The authors are convinced that the reason for the increase in alcohol consumption is the abandonment of state monopoly, characteristic for communist countries. However, they do not address the reasons for the increase in alcohol consumption other than its free market availability and promotion. One could, however, ask about non-market determinants of the growing alcohol consumption, e.g. cultural, political, psychological, etc. This means that they associate the possible decrease in alcohol consumption only with an increase in taxes, an expansion of legal restrictions, and increased state control and monitoring. This does not seem to be a legitimate conclusion.

We acknowledge the long cultural history of alcohol use in China in both the abstract and the text. This is well covered in the existing literature. The supply of alcohol in China is less well covered in the literature and this is the reason for the focus of the paper.

The effectiveness of the policies reviewed in the paper are well established as effective in the alcohol policy literature.

The authors report that according to the World Health Organization, per capita alcohol consumption in China in 2016 was 12.5% higher than the global average (142-143), and then go on to state that it is lower than in many highly developed countries (283-285). This would seem to require additional commentary.

These comments are not contradictory. In many countries very little alcohol is consumed (a majority of the world are abstainers) and so China is above the global average while being lower than many high-income countries. These data are available via the World Health Organization.

Reviewer 3 Report

Dear authors, thank you for sending the manuscript. After reviewing it, you proceeded to indicate certain important aspects that must be modified in order to reconsider publication. • The method is not adequately explained, they do not explain the type/design of the study, neither search limiters nor inclusion/exclusion criteria…. • Does not detail how the information was collected using the snowball technique. • There would be a lack of a scheme detailing how the selection process of reviewed articles / selected articles has been. • It would be necessary for the authors to expose the limitations of the study • References must be reviewed and placed according to the regulations of the journal • The wording of the conclusions section must be rewritten, since it is written as if it were a discussion, these do not respond to the objectives set. • The authors must justify more those more general aspects of the introduction Greetings

Author Response

Comments and Suggestions for Authors

Dear authors, thank you for sending the manuscript. After reviewing it, you proceeded to indicate certain important aspects that must be modified in order to reconsider publication. • The method is not adequately explained, they do not explain the type/design of the study, neither search limiters nor inclusion/exclusion criteria…. •

The design is described as a narrative review or topical approach designed alongside an analysis of official documents to track China’s changing alcohol market and need for enhanced policy response. Narrative, or non-systematic reviews, do not usually attempt to locate all relevant literature. A narrative review tracks the development topics requiring wider scoping, the narrative thread of which may be lost in the restrictive rules of systematic review. “Strengths of narrative review include consolidation of previous work, summation, identification of omissions or gaps, and achieving new insights from identifying previously unknown, nonobvious connections, thus developing fresh conceptions” (Chaney, 2021)*. 

*Chaney, M. (2021). So you want to write a narrative review article? [Editorial]. Journal of Cardiothoracic and Vascular Anesthesia, 35, 3045-49.

 Does not detail how the information was collected using the snowball technique. •

Agreed, snowball is usually used to refer to sampling of participants. We have replaced with ‘forward/backward citation searches’

There would be a lack of a scheme detailing how the selection process of reviewed articles / selected articles has been.

This was covered in the methods section where the key words and process of evaluation by the researchers is outlined.

  • It would be necessary for the authors to expose the limitations of the study

We have included as Limitations: ‘Narrative reviews are not systematic so some relevant materials could have been missed but the authors’ goal was to provide a narrative thread on the topic rather than a comprehensive, catch-all review. A second limitation is that only official information that is publicly available could be accessed.’

  • References must be reviewed and placed according to the regulations of the journal

The references have been formatted according to the guidelines found here: https://www.mdpi.com/journal/ijerph/instructions#references

  • The wording of the conclusions section must be rewritten, since it is written as if it were a discussion, these do not respond to the objectives set.

Thank you, this has been rewritten.

  • The authors must justify more those more general aspects of the introduction

Not clear what this means?

Round 2

Reviewer 3 Report

Dear Authors, thank you for attending and justifying the comments and suggestions for improvement. After a second revision, my decision is to accept the manuscript with minor revision. In this sense, the authors must review the bibliographic references and put them according to the guidelines set by the journal. Greetings

Author Response

Dear Authors, thank you for attending and justifying the comments and suggestions for improvement. After a second revision, my decision is to accept the manuscript with minor revision. In this sense, the authors must review the bibliographic references and put them according to the guidelines set by the journal.

Thank you for the nice decision, we revised the references according to the guidelines of the journal.